# Productive and Environmental Consequences of Sixteen Years of Unbalanced Fertilization with Nitrogen and Phosphorus—Trials in Poland with Oilseed Rape, Wheat, Maize and Barley

**Agnieszka Rutkowska \*** and **Piotr Skowron**

Department of Plant Nutrition and Fertilization, Institute of Soil Science and Plant Cultivation—State Research Institute, Czartoryskich 8, 24-100 Puławy, Poland; pskowron@iung.pulawy.pl
\* Correspondence: agrut@iung.pulawy.pl; Tel.: +48-81-4786840

**Abstract:** Two factorial field experiments were carried out between 2003 and 2018 in the Experimental Stations in Eastern and Western Poland using four crop rotations with winter oilseed rape, winter wheat, maize and spring barley. The initial value of phosphorus (P) in Grabów soil was 69.8 mg P·kg$^{-1}$ soil and in Baborówko soil it was 111.3 mg P·kg$^{-1}$ soil (Egner-Riehm Double-Lactate DL). P fertilizer was added annually at 39 kg P·ha$^{-1}$ under winter oilseed rape, 35 kg P·ha$^{-1}$ under maize and 31 kg P·ha$^{-1}$ under wheat and barley using superphosphate and nitrogen (N), which was added at five levels (30–250 kg N·ha$^{-1}$) per year as ammonium nitrate in addition to controls with no added fertilizer. Through the several years of the experiment, P fertilizer had no effect on crop N use efficiency (NUE) nor crop productivity. There was significant soil P mining particularly in the high-N fertilizer trials causing a reduction in the content of available soil P by up to 35%. This work recommends that, based on soil P analysis, P fertilizer should not be added to high-P soils. This practice may continue uninterrupted for several years (16 in this case) until the excess soil P has been reduced. This mechanism of removal of "legacy" P from soil has major implications in reducing runoff P into the Baltic Sea drainage area and other water bodies.

**Keywords:** crop productivity; nitrogen use efficiency; nutrient balance; P use efficiency; plant available P

## 1. Introduction

Nitrogen (N) and phosphorus (P) are the two most essential nutrients ensuring food production. Both N and P are involved in vital plant functions such as photosynthesis protein formation, and symbiotic N fixation [1]. Simultaneously, N is the most important crop yield-limiting factor in the world [2]. As a main nutrient element, it is needed in a relatively large quantity for the production of proteins, nucleic acids and chlorophyll in plants [3]. The natural sources of N and P are very different: in principle, N availability is unlimited as an atmospheric gas, whereas P comes from rock phosphate, renewed with the uplift of continental rock [4]. In ecosystems, N and P are bound within waste organic products and dead organic matter and must be remineralized to release inorganic orthophosphates or dissolved and reduced to inorganic nitrate and ammonia before either element can be absorbed by plants.

However, since the 1960s, anthropogenic activities have dramatically modified the cycle of N and P in the biosphere, threatening natural ecosystems [4]. The invention of the Haber–Bosch process more than 100 years ago, allowed for the production of synthetic mineral fertilizers from ammonia on a large scale [5]. Thus, European N fertilizer use has grown from about 1.2 Tg in 1961 to 11.8 Tg in 2018 [6].

The increased use of N fertilizers has contributed greatly to the increased global food, feed and biofuel production [7]. Nevertheless, the great availability of N in agriculture increases losses to the wider environment, which can lead to problems related to human health and ecosystem degradation [8–10]. The volatilization of ammonia, leaching nitrate, and the emissions of di-nitrogen, nitrous oxide and nitrogen oxide are the main loss pathways of agricultural systems [11].

Numerous studies show that less than 50% N and 25% P introduced into agroecosystems in the form of mineral and natural fertilizers are effectively used while the rest is dispersed in the environment, and contributes to various negative ecological effects [12–14]. Nitrogen use efficiency (NUE) is considered quite low on average in conventional agricultural systems in the world, including developed countries. The NUE for world cereal production has been estimated for 33% of fertilizer N recovered by the crop [15].

The excessive eutrophication of waters, especially in marine coastal areas and estuaries, has been a growing problem worldwide for many decades. It is caused both on a global and local scale by the man-made intensification of biogeochemical cycles of N and P, which increases the loads of nutrients, introduced into the water from point and diffuse sources [16–21]. The main streams of nutrients, apart from natural sources, come from food consumption (point sources) and agriculture (diffuse nutrient outflow), which are broadly understood, where the significant increase in the N and P load is related to unbalanced mineral fertilization and the intensification of animal production [13,21].

To reduce eutrophication in the Baltic Sea, the HELCOM Contracting Parties agreed in 2007 and 2013 on the Baltic Sea Action Plan (BSAP), one of the main elements of which is the Nutrient Input Reduction Scheme, where the reduction targets are divided among the coastal countries [22,23]. The revised (2013) waterborne inputs for the entire Baltic Sea were determined as annual reductions of 89.26 t N and 14.37 t P [21,24]. The ecological state of the Baltic Sea is systematically improving as a result of international efforts, but according to HELCOM, excessive eutrophication still occurs in most parts of the basin [25].

The risk of P loss increases with its concentration in soil [26] and within 21 days of a fresh application of highly water-soluble P fertilizers [27]. The addition of P as reactive phosphate rocks has been shown to decrease P losses compared to superphosphate by 10–80% at the plot, field and catchment scale [28], which is caused by a combination of low water solubility and a greater density of fertilizer particles, reducing the availability of P loss [29].

On the other hand, the mining of phosphate rocks for use in agricultural fertilizers has increased dramatically in the latter half on the past century. In the worst-case scenarios, about 40–60% of the current resource base will be extracted by 2100 [30]; under the best estimates, the depletion would be around 20–35% in the time. The environmental implications and the finite character of rock phosphate make it necessary to reconsider the views on this element in crop production.

Recent studies have suggested that the amount of reactive N compounds emitted into the environment is far too high and already exceeds the "safe operating space for humanity". More appropriate management of N use is therefore of key importance, but it should be emphasized that the proper management of the element is linked to the balanced fertilization with other nutrients to achieve both economic and environmental goals [31,32]. The essential, irreplaceable role of P in plant nutrition requires that sufficient P is available in soil to meet the need of the growing crop in order to produce optimum, economic yields. The need varies through the life cycle of the plant and is usually greatest in the early stages of growth, when the root system is not fully developed, to explore the maximum volume of soil [33,34]. It has been found that, under insufficient P availability, crops do not respond to N fertilization [35,36]. There is evidence that high yielding crop varieties need more N and P to realize their potential and thus under insufficient fertilization with P the reserve of P in soil might be exhausted and lead to yield reduction. Nevertheless, an important issue is to recognize the possibility of reduced P fertilizer use on soils with a high concentration of available forms of the element.

This paper shows that the amount of P applied in agriculture could be considerably reduced by improving the fertilizer recommendation. In the data collected between 2003 and 2018, including four complete rotations of the most popular crops in Poland (winter oilseed rape, winter wheat, maize for corn and spring barley) fertilized with increasing N rates, the impact of long-term P cessation in the soil on crop productivity and changes of the available forms of P have been described.

The following hypothesis has been set: the cessation of P fertilization on soils with a very high and high available P content does not impact crop N use efficiency, and neither does it cause excessive soil P depletion in the long term.

## 2. Materials and Methods

The long-term field experiments were carried out between 2003 and 2018 at the Experimental Stations of the Institute of Soil Science and Plant Cultivation in Grabów (21°39′ E, 51°21′ N) and Baborówko (16°37′ E, 52°37′ N), Poland. Both Experimental Stations are located at the cold temperate dry climate zone in which potential evapotranspiration normally exceeds rainfall during the whole vegetation period. Average annual rainfall at Baborówko was 492 mm and at Grabów it was 587 mm during the period of investigation. In Experimental Station Grabów, soil under the experiment was heterogeneous—sandy loam (World Reference Base WRB: Stagnic Luvisols). In Baborówko the experiment was localized partly on the sandy loam (WRB: Albic Luvisol) and partly on the black earth (WRB: Gleyic Phaeozem) [37]. In two factorial, two replication experiments, four crops were grown each year in the rotation of: winter oilseed rape (*Brassica napus* L. *var. oleifersa*); winter wheat (*Triticum aestivum* L.); maize (*Zea mays* L.); spring barley (*Hordeum vulgare* L.). In Baborówko during 2015–2018 and in Grabów in 2015 and 2018, mustard (*Sinapis alba* L.) was grown instead of winter oilseed rape because of the extremely unfavorable conditions for the emergence or wintering of the crop.

During the experiment, no manure was applied, no leguminous plants were grown and straw was removed from the field. In the split-plot layout, the first variable was P fertilizer in the P-plus and P-minus (control) treatments and the second variable was six levels of N fertilizer, including a control with no N supply. In Grabów, the soil was characterized as slightly acid ($pH_{KCl}$ 6.2) (according to the classification of soil reaction used in Poland, based on the measurement of pH in 1M KCl 1:5, soil:solution), had a medium content of available potassium (88.8 mg·kg$^{-1}$ soil) (Egner-Riehm Double-Lactate DL method) and a high content of available magnesium (30.4 mg Mg·kg$^{-1}$ soil) (Schachthabel method). In Baborówko, the soil reaction was neutral ($pH_{KCl}$ 6.8), the content of potassium was high (116 mg K·kg$^{-1}$ soil) and the concentration of magnesium was very high (54 mg Mg·kg$^{-1}$ soil).

In Grabów, the initial content of available P in 2003 was recognized as high (69.8 mg P·kg$^{-1}$ soil) and in Baborówko as very high (111.3 mg P·kg$^{-1}$ soil), according to the classes of available P content in soil used in the Polish recommendation system. Since 2003, in the P-plus treatment, P fertilizer as a superphosphate was applied before sowing crops at the following rates: 39 kg P·ha$^{-1}$·y$^{-1}$ under winter oilseed rape, 35 kg P·ha$^{-1}$·y$^{-1}$ under maize and 31 kg P·ha$^{-1}$·y$^{-1}$·under wheat and barley. In the control, no P fertilizer was added past 2003. Potassium (66.0–116 kg K·ha$^{-1}$·y$^{-1}$) and magnesium (42 kg Mg·ha$^{-1}$·y$^{-1}$) were constant in the P-plus and in the control. Nitrogen fertilizers were applied as ammonium nitrate. Spring barley was fertilized with 0, 30, 60, 90, 120, 150 kg N·ha$^{-1}$·y$^{-1}$, winter wheat with 0, 40, 80, 120, 160, 200 kg N·ha$^{-1}$·y$^{-1}$, and winter oilseed rape and maize with 0, 50, 100, 150, 200 and 250 kg N·ha$^{-1}$·y$^{-1}$. For spring barley, the first dose of 30 kg N·ha$^{-1}$·y$^{-1}$ was applied at sowing, and the successive N doses were carried out over two-week intervals. For winter wheat, the first dose of 40 kg N·ha$^{-1}$·y$^{-1}$ was applied at the beginning of the spring vegetation, and the successive N doses were applied as for barley. For maize and winter oilseed rape, the total rates of 50 and 100 kg N·ha$^{-1}$·y$^{-1}$ were applied after emergence and in the early spring, respectively. In the N150-N250 treatments, the first rate of 100 kg N·ha$^{-1}$·y$^{-1}$ was applied after emergence and in the early spring, similarly, and the subsequent additions of 50 kg N·ha$^{-1}$·y$^{-1}$ were spread over 14-day intervals.



Soil samples from the plough layer were collected each year between 2007 and 2018 after the harvest of barley. The samples were dried, passed through a 2-mm sieve and analyzed for available P and potassium according to the Egner-Riehm DL method [38] with UV–Vis spectrophotometry (Nicolet Evolution 300), and for available magnesium according to the Schachtschabel method [39] with F-AAS spectroscopy (Varian Spectra AA-240 FS).

The grain and straw yields of winter oilseed rape, winter wheat and spring barley were harvested by a plot harvester; maize was harvested by hand. Wheat and barley were harvested over an area of 17.9 m$^2$ and rape and maize over 19.1 and 20.3 m$^2$, respectively. The grain yields of wheat, barley and corn were determined at a moisture level of 15%, and winter oilseed rape was determined at 12%. The plant samples of grain and straw for each crop were collected at full maturity from an area of 1 m$^2$ and analyzed for total N and P as determined by the mineralization of the sample using the Kjeldahl method with Continuous Flow Analysis (CFA) and UV–Vis spectrophotometry (Skalar San+). The uptake of N and P in grain and straw was calculated from the percentage of the elements and the yields of the main and fore crop.

Following the method proposed by the EU Nitrogen Expert Panel (NEP), N use efficiency (NUE) was calculated for the crops to determine the effect of plants' P supply on this parameter. According to the approach, NUE calculations based on N input and N output at different levels provide information about resource use efficiency, the economy of food production (N in harvested yield) and the pressure on the environment (N surplus). The desired value of NUE, N uptake (Yn) and N surplus (Ns) should be in the range NUE 50–90%, Yn > 80 kg N·ha$^{-1}$ and Ns < 80 kg N·ha$^{-1}$ [40]. Nitrogen use efficiency was calculated according to the formula [40]:

$$\text{NUE} = \frac{\text{Yn}}{\text{F}} \times 100$$

where Yn is total N uptake by the crop (kg·ha$^{-1}$); F is N fertilizer rate (kg·ha$^{-1}$).

N surplus (Ns) was calculated according to the formula [40]:

$$\text{Ns} = \text{F} - \text{Yn}$$

P use efficiency (PUE) (%) by the crops in the P-plus treatment was calculated according to the formula [41]:

$$\text{PUE}(\%) = \frac{\text{Pnplus} - \text{Pncontrol}}{\text{PF}} * 100$$

where Pn plus is total P uptake by plants fertilized with P (kg P·ha$^{-1}$), Pn control is total P uptake by plants without added P fertilizer (kg P·ha$^{-1}$) and PF is the P fertilizer rate (kg·ha$^{-1}$).

P surplus (Ps) was calculated according to the formula [41]:

$$\text{Ps} = \text{PF} - \text{Pn}$$

The data, derived over the sixteen years, were processed using an analysis of variance (ANOVA). Tukey's test was applied to evaluate the significance of differences between the treatments. Statistical processing of the results were performed using the Statgraphics Centurion 18 Package (Statgraphics Centurion, Rockville, MA, USA).

## 3. Results

### 3.1. Effect of P Additions on Crop Productivity

The grain yields derived from the 2003–2018 P trials averaged over the six levels of N fertilizer are presented in Figures 1–4. The general trend is that there were no significant increases in the growth of these crops due to added P. In general, higher yields were obtained in Grabów than in Baborówko.

Within the analyzed period, comprising several years of experiments, there was found to be significant soil P mining by the crops in the controls where P fertilizer was not added.

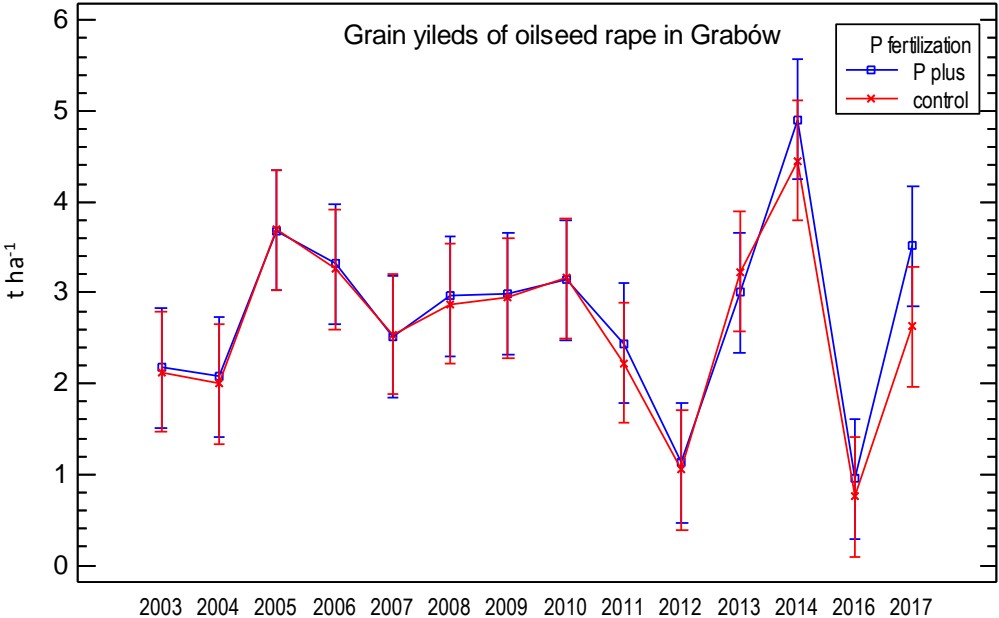

(a)

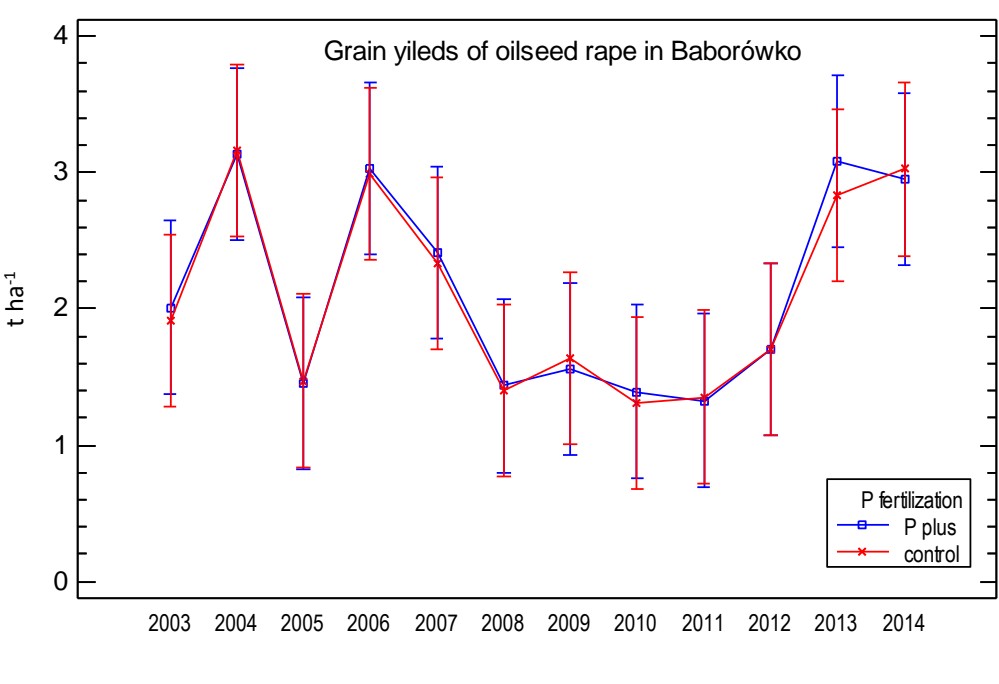

(b)

**Figure 1.** Grain yields of winter oilseed rape in (**a**) Grabów and (**b**) Baborówko; 95% Tukey Honest Significant difference test (HSD) intervals; $n$ = 336 for Grabów and $n$ = 288 for Baborówko.

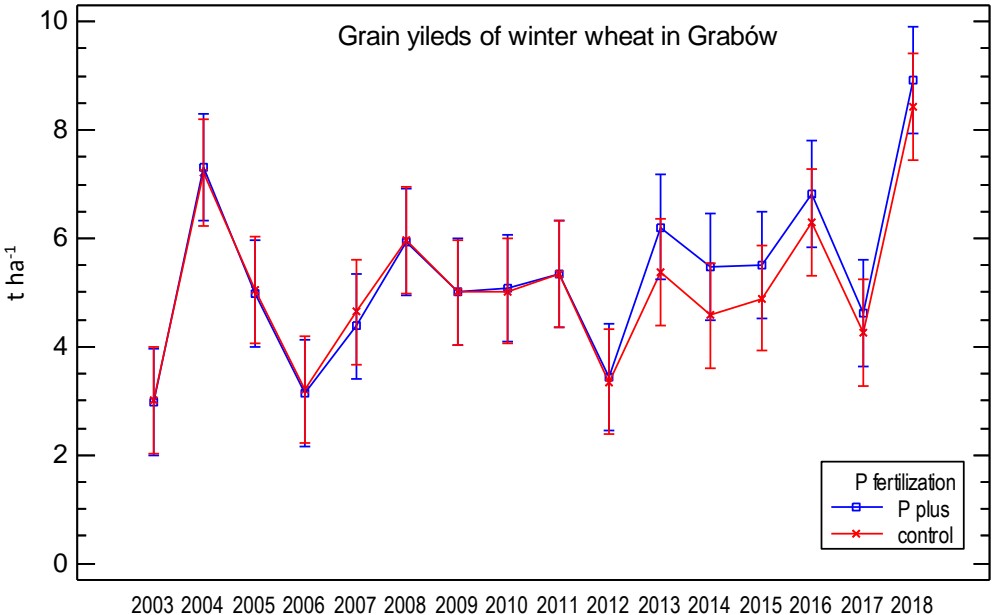

(a)

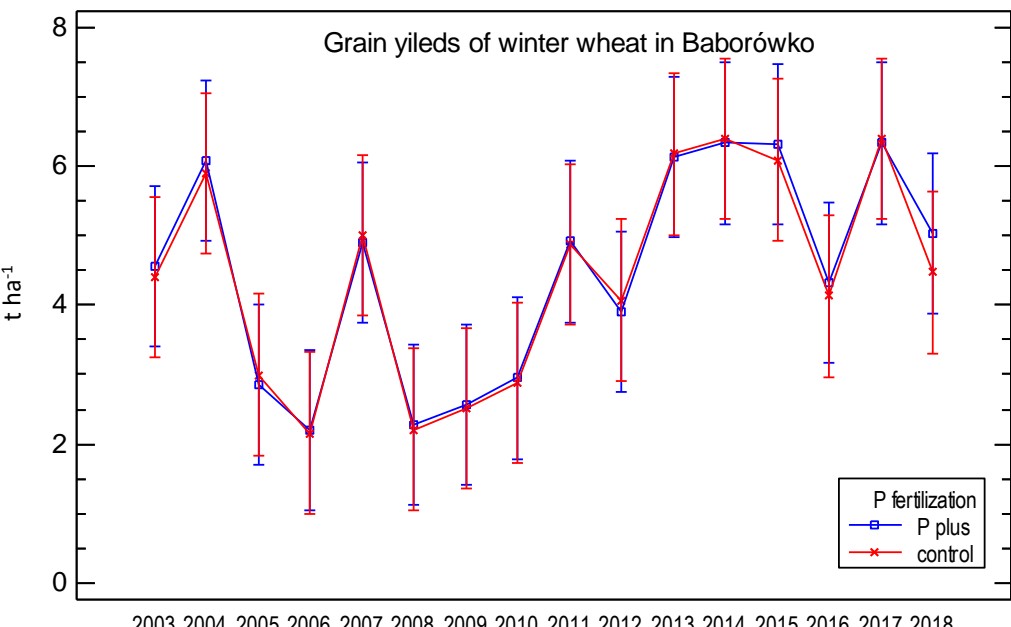

(b)

**Figure 2.** Grain yields of winter wheat in (**a**) Grabów and (**b**) Baborówko; 95% Tukey HSD intervals; *n* = 384 for each location.

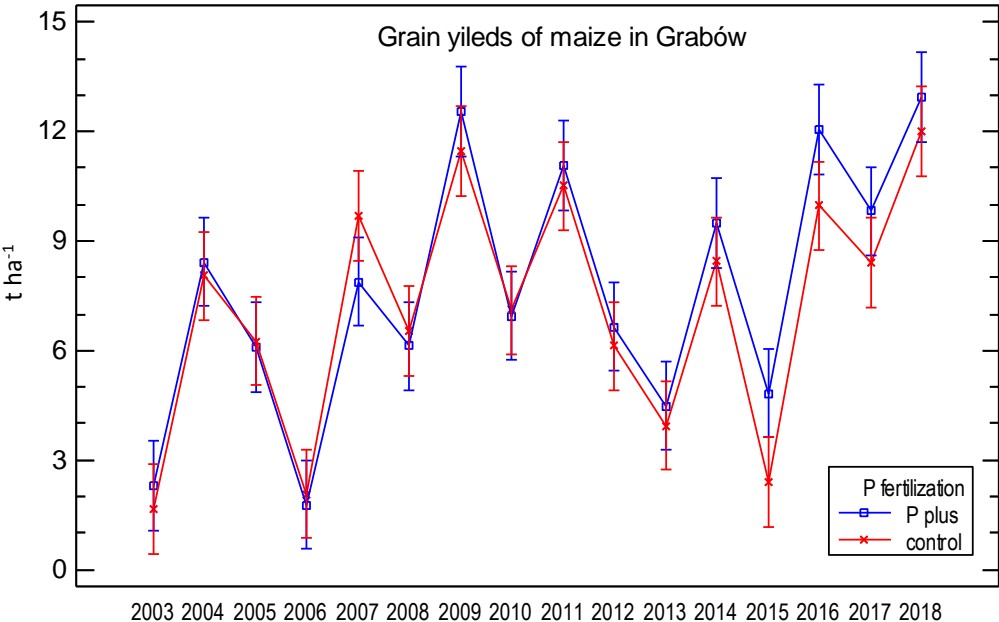

(a)

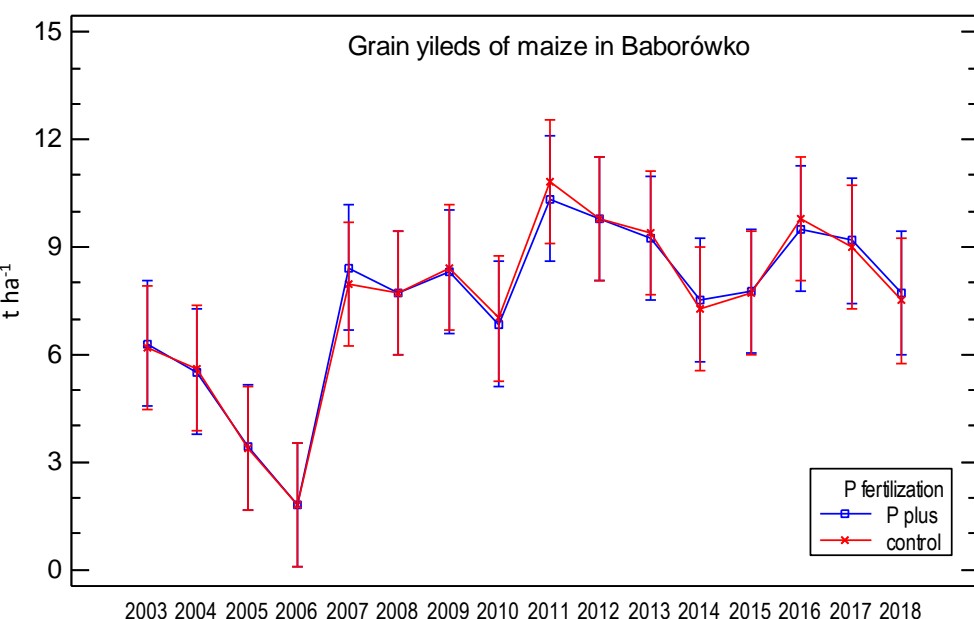

(b)

**Figure 3.** Grain yields of maize in (**a**) Grabów and (**b**) Baborówko; 95% Tukey HSD intervals; *n* = 384 for each location.

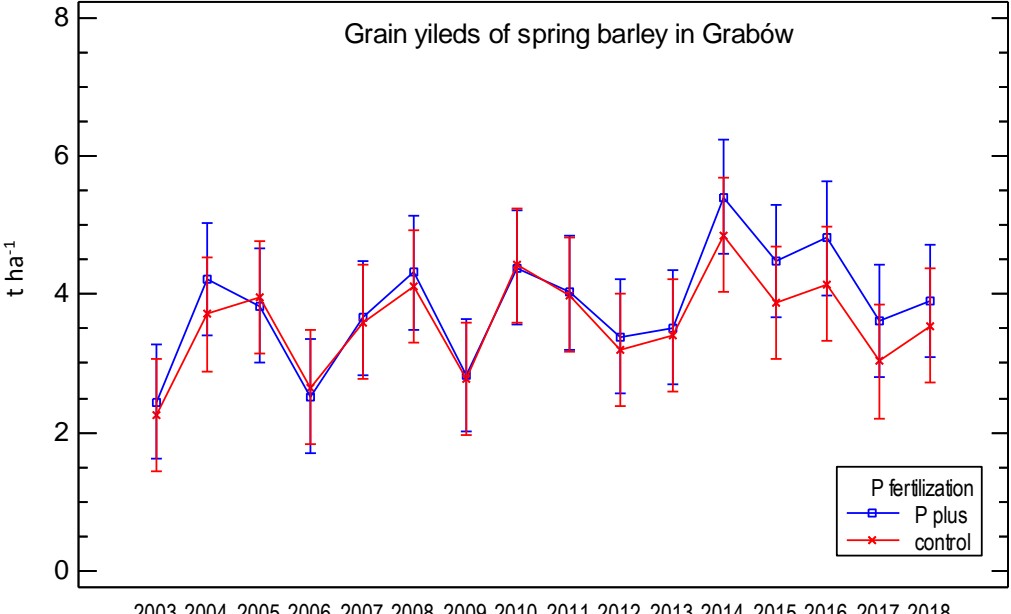

(a)

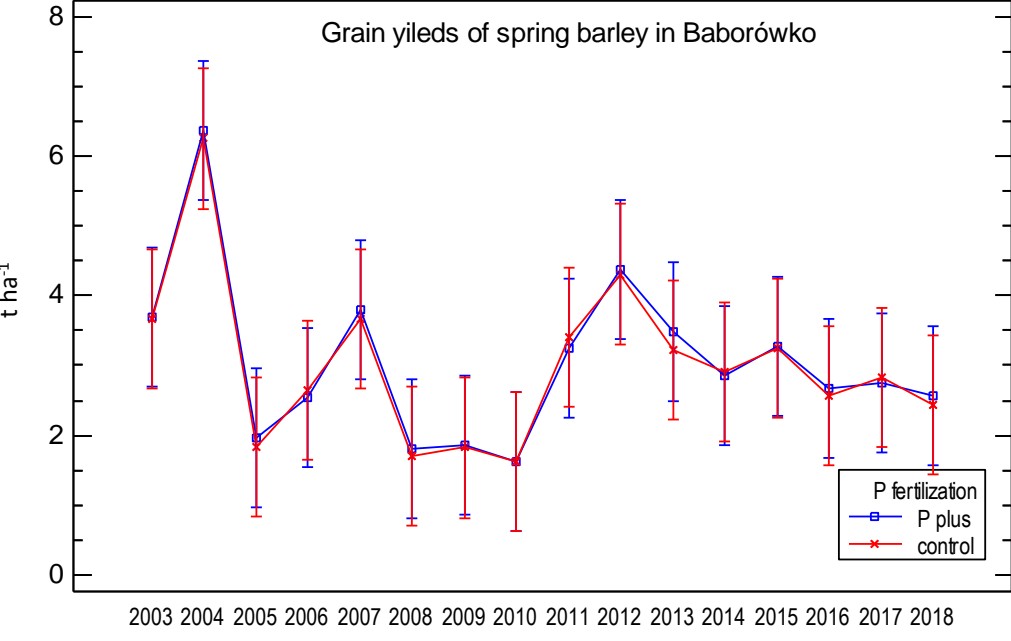

(b)

**Figure 4.** Grain yields of spring barley in (**a**) Grabów and (**b**) Baborówko; 95% Tukey HSD intervals; *n* = 384 for each location.

The yield of winter oilseed rape oscillated between 0.85 and 4.68 t·ha$^{-1}$ in Grabów, and in Baborówko it ranged from 1.34 to 3.15 t·ha$^{-1}$ (Figure 1). In Grabów, the average yields obtained in the P-plus treatment were 2.77 and 2.63 t·ha$^{-1}$ in the control. In 2017, when the crop was grown for

the last time, a yield reduction of 0.85 t·ha$^{-1}$ was observed in the treatment where P fertilization had ceased (Figure 1a). In Baborówko, the seed yields were comparable between the treatment and control achieving an average of 2.1 t·ha$^{-1}$ (Figure 1b).

In Grabów, the average grain yields of winter wheat were 5.32 t·ha$^{-1}$ in P-plus and 5.10 t·ha$^{-1}$ in the control. In Baborówko, the mean grain yields were 4.40 t·ha$^{-1}$ for both P-plus and the control (Figure 2).

In Grabów, the yields of maize showed a wide range from 1.9 to 12.4 t·ha$^{-1}$. During 2015–2018, the crop yield was slightly lower in the control compared to the P-plus treatment. (Figure 3a). In the same years, maize in Baborówko yielded levels of 8.80 t·ha$^{-1}$ and no differences between P-plus and the control were noted (Figure 3b).

The yields of spring barley in Grabów were more stable compared to the other crops and ranged from 2.5 to 5.1 t·ha$^{-1}$ (Figure 4a). Similar to maize in this location during 2014–2018, barley growth was lower in the control. In Baborówko, the yields of barley were variable, with no P fertilizer effect (Figure 4b).

### 3.2. Crop Response to the Combined N and P Fertilizer Additions

The average grain yields for 2003–2016 of winter oilseed rape, winter wheat, maize and spring barley are presented in Figures 5–8. The crop response to added N showed clear positive trends. In Grabów, for both P treatments, the increased yields of winter oilseed rape were statistically significant at 150 kg N·ha$^{-1}$, winter wheat at 120 kg N·ha$^{-1}$, maize at 100 kg N·ha$^{-1}$ and barley at 120 kg N·ha$^{-1}$. In Baborówko, crop growth increased with increasing N fertilizer additions, except for maize, which did not respond to N fertilization above 150 kg N·ha$^{-1}$, regardless of the level of added P.

In Grabów, a tendency for grain yield reduction together with N rates was found, in particular for maize; however, the interaction between the two experimental factors has not been statistically proven. The average grain yields of maize in the P-plus treatment ranged from 5.90 to 8.36 t·ha$^{-1}$ and in the P control between 5.40 and 8.07 t·ha$^{-1}$, depending on the level of N fertilization (Figure 7). There was no similar relationship observed in Baborówko, where the average yields achieved in the analysis period were practically the same, regardless of the P supply.

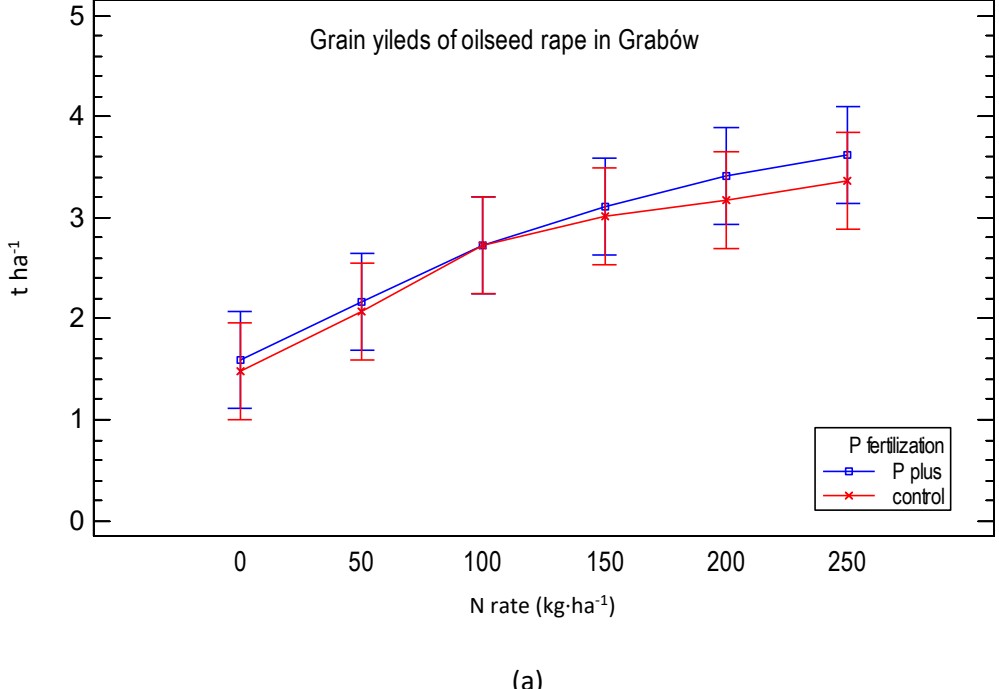

(a)

**Figure 5.** *Cont.*

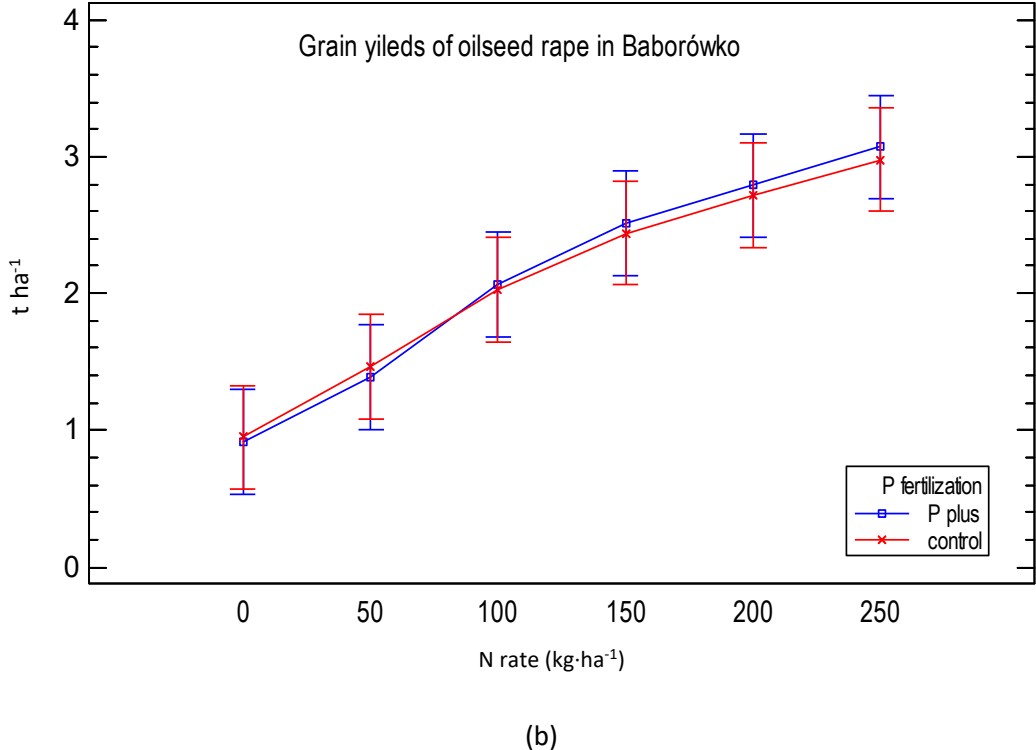

(b)

**Figure 5.** Grain yields of winter oilseed rape at increasing additions of Nitrogen (N) fertilizer at the P-plus and control levels of phosphorus (P); 95% Tukey HSD intervals; *n* = 336 for (**a**) Grabów (P fertilization HSD 0.405) and *n* = 288 for (**b**) Baborówko (P fertilization HSD 0.321).

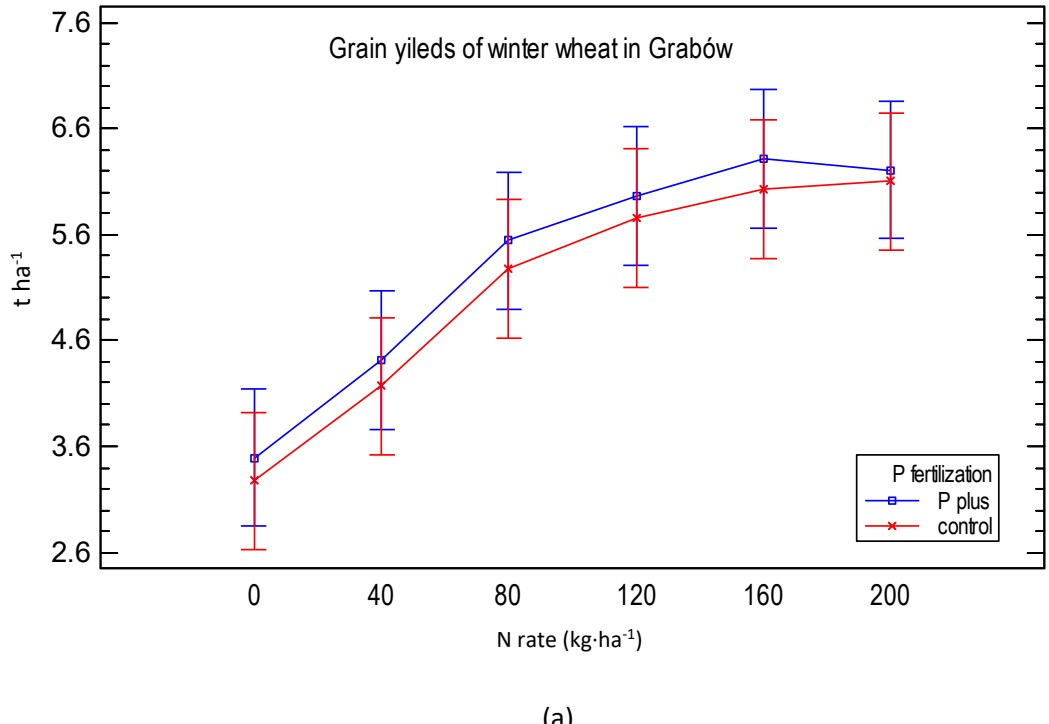

(a)

**Figure 6.** *Cont.*

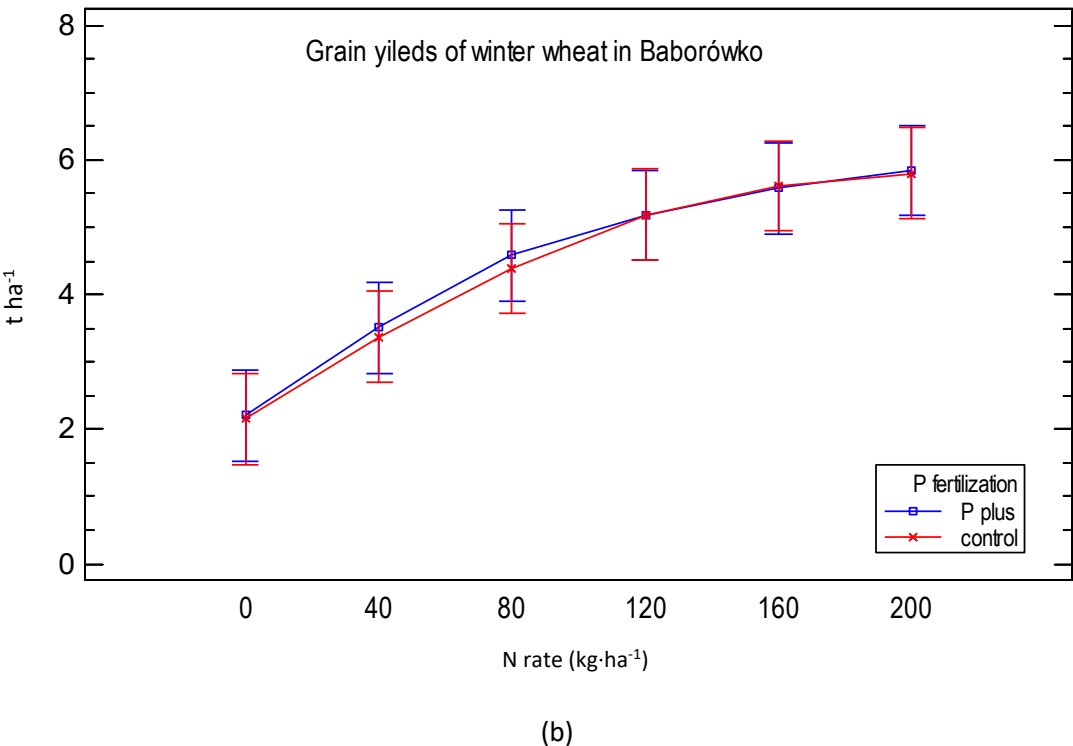

(b)

**Figure 6.** Grain yields of winter wheat at increasing additions of N fertilizer at the P-plus and control levels of P; 95% Tukey HSD intervals; *n* = 384 for each location; (P fertilization (**a**) Grabów HSD 0.550, (**b**) Baborówko HSD 0.568).

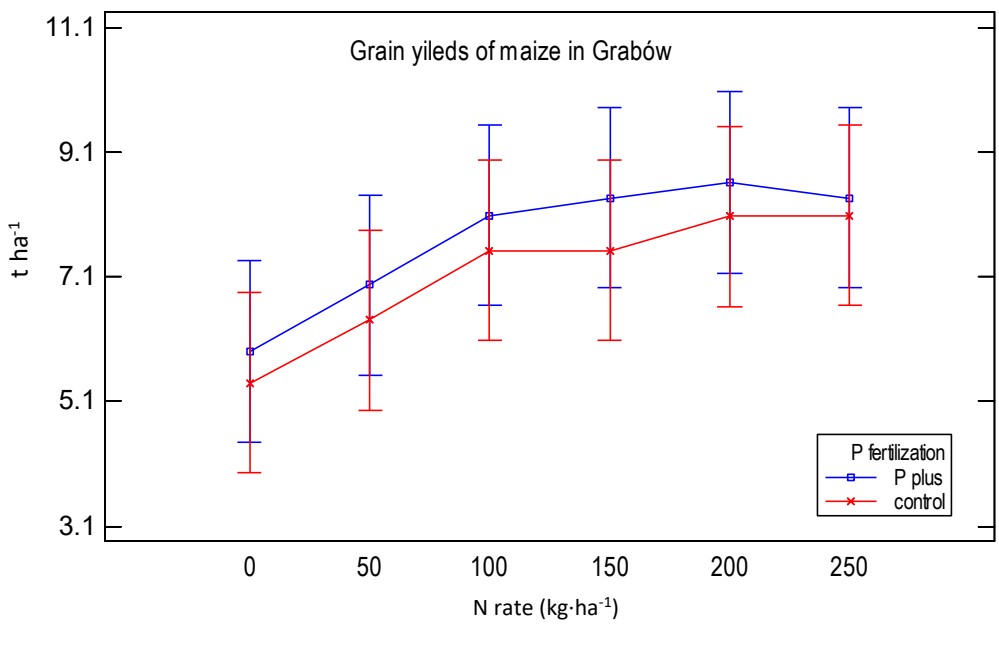

(a)

**Figure 7.** *Cont.*

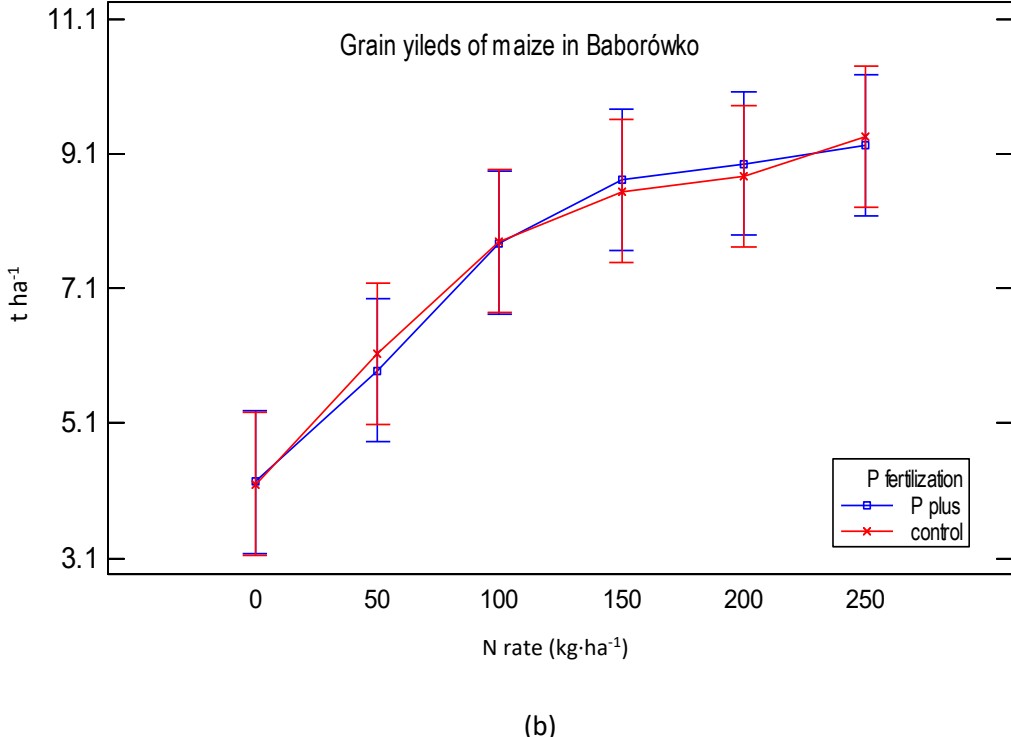

(b)

**Figure 7.** Grain yields of maize at increasing additions of N fertilizer at the P-plus and control levels of P; 95% Tukey HSD intervals; *n* = 384 for each location (P fertilization (**a**) Grabów HSD 1.183, (**b**) Baborówko HSD 0.894).

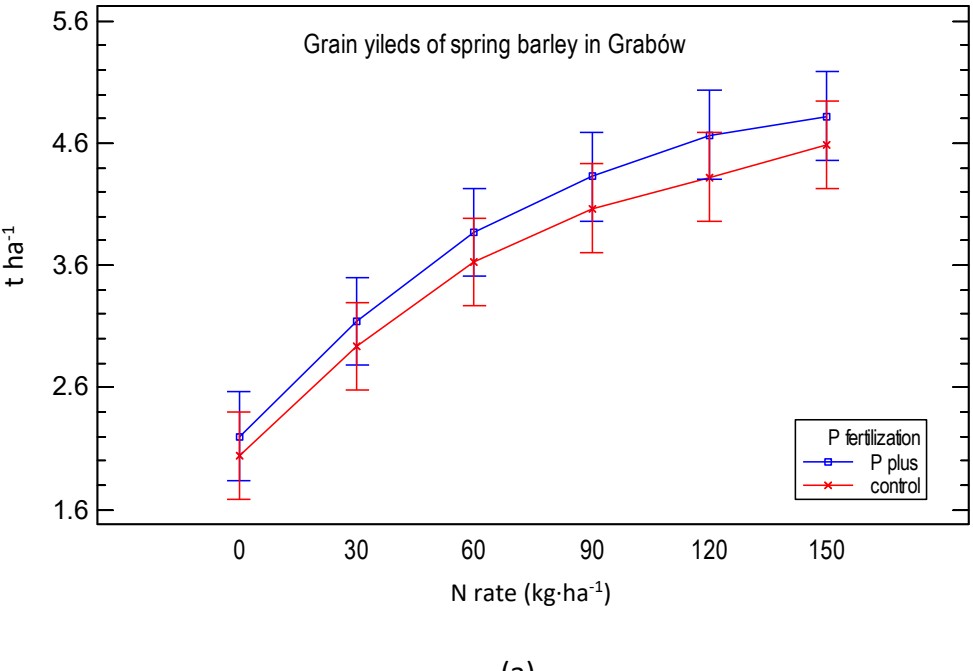

(a)

**Figure 8.** *Cont.*

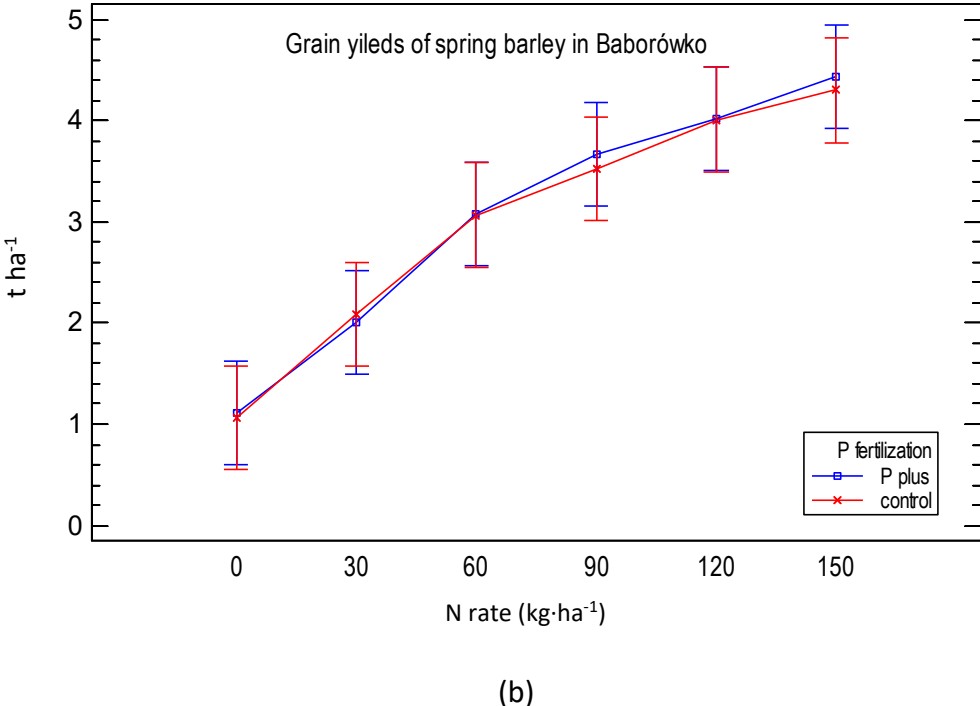

(b)

**Figure 8.** Grain yields of spring barley at increasing additions of N fertilizer at the P-plus and control levels of P, 95% Tukey HSD intervals; *n* = 384 for each location (P fertilization (**a**) Grabów HSD 0.349, (**b**) Baborówko HSD 0.434).

### 3.3. Effects of N Additions on P Uptake and Use Efficiency

The following indices characterizing P use efficiency were calculated: P uptake (Pn), P surplus (Ps) and P use efficiency (PUE) by plants fertilized with P. The effect of N additions on these indices are presented in Tables 1–4. In Grabów there were significant effects of P additions on P uptake, but no additional effect of N fertilization was seen. In Baborówko, P uptake by the crops was not affected by P additions.

The P uptake by winter oilseed rape, wheat, maize and barley increased together with N rates. The average P uptake by winter oilseed rape was 21 kg P·ha$^{-1}$ in Grabów and 15 kg P·ha$^{-1}$ in Baborówko (Table 1). The uptake of P in wheat reached 20 and 17 kg P·ha$^{-1}$ (Table 2), in maize 29 and 24 kg P·ha$^{-1}$ (Table 3) and in barley 20 and 15 kg P·ha$^{-1}$ (Table 4), respectively.

In the treatment with P fertilizer, the surplus P calculated for the crops was positive over the whole range of N doses. The highest value of surplus P was found in the winter oilseed rape trial (18 kg P·ha$^{-1}$ in Grabów and 24 kg P·ha$^{-1}$ in Baborówko, on average) and the lowest was in the maize trial (7.0 kg P·ha$^{-1}$ in Grabów and 10 kg P·ha$^{-1}$ in Baborówko). Where no P fertilizer was added since 2003, the surplus P was negative with increasing negative values governed by the rate of N added. The most negative P surplus was obtained in the maize trials in both locations (−27 kg P·ha$^{-1}$ in Grabów and −23 kg P·ha$^{-1}$ in Baborówko on average) as a consequence of the enhanced P uptake by the crop. The P use efficiency (PUE) in the experiments was low, averaging 3–7%.

**Table 1.** P use indices for winter oilseed rape; N fertilizer rate; P fertilizer rate; Pn (uptake) and Ps (surplus) (kg·ha⁻¹); P use efficiency (PUE) (%).

| N Rate | Grabów | | | | | Baborówko | | | | |
|---|---|---|---|---|---|---|---|---|---|---|
| | P-Plus 39 kg P·ha⁻¹ | | | Control 0 kg P·ha⁻¹ | | P-Plus 39 kg P·ha⁻¹ | | | Control 0 kg P·ha⁻¹ | |
| | PUE | Pn | Ps | Pn | Ps | PUE | Pn | Ps | Pn | Ps |
| 0 | 3 [a] | 13.2 [aA] | 26 [aA] | 12.0 [aA] | −12 [a] | 0 | 6.91 [aA] | 32 [a] | 6.87 [aA] | −7 [aB] |
| 50 | 3 [a] | 17.2 [bA] | 22 [bA] | 16.1 [bA] | −16 [b] | 0 | 9.99 [bA] | 29 [b] | 10.6 [bA] | −11 [bB] |
| 100 | 4 [ab] | 21.0 [cA] | 18 [cA] | 19.4 [cA] | −19 [c] | 4 [ab] | 14.9 [cA] | 24 [c] | 13.2 [cA] | −13 [cB] |
| 150 | 4 [ab] | 23.0 [dA] | 16 [dA] | 21.3 [cdA] | −21 [cd] | 3 [a] | 17.8 [dA] | 21 [d] | 16.6 [dA] | −16 [dB] |
| 200 | 7 [b] | 24.5 [eA] | 15 [deA] | 21.6 [cdB] | −22 [cd] | 5 [bc] | 19.1 [eA] | 20 [e] | 17.2 [eA] | −17d [eB] |
| 250 | 12 [c] | 26.5 [fA] | 13 [fA] | 22.5 [dB] | −23 [d] | 5 [ab] | 21.2 [fA] | 18 [f] | 19.3 [fA] | −19 [fB] |
| **Mean** | **6** | **21** | **18** | **19** | **−18** | **6** | **15** | **24** | **14** | **−14** |

Treatments with the same letter are not significantly different (*p* ≤ 0.05). The capital letters denote differences between the mean of Pn, Ps in P-plus and control treatments and the small letters denote difference between N rates; *n* = 280 for Grabów; *n* = 240 for Baborówko.

**Table 2.** P use indices for winter wheat; N fertilizer rate; P fertilizer rate; Pn (uptake) and Ps (surplus) (kg·ha⁻¹); PUE (%).

| N Rate | Grabów | | | | | Baborówko | | | | |
|---|---|---|---|---|---|---|---|---|---|---|
| | P-Plus 31 kgP·ha⁻¹ | | | Control 0 kgP·ha⁻¹ | | P-Plus 31 kgP·ha⁻¹ | | | Control 0 kgP·ha⁻¹ | |
| | PUE | Pn | Ps | Pn | Ps | PUE | Pn | Ps | Pn | Ps |
| 0 | 5 [ab] | 15.0 [aA] | 16 [aA] | 13.5 [a] | −14 [aB] | - | 8.85 [aA] | 22 [aA] | 9.00 [aA] | −9 [aB] |
| 40 | 6 [bc] | 17.3 [bA] | 14 [bA] | 15.5 [bB] | −16 [bB] | - | 13.0 [bA] | 18 [bA] | 13.9 [bA] | −14 [bB] |
| 80 | 4 [a] | 19.5 [cA] | 12 [cA] | 18.3 [cA] | −18 [cB] | 4 [a] | 17.5 [cA] | 14 [cA] | 16.4 [cA] | −16 [cB] |
| 120 | 4 [a] | 21.8 [dA] | 9 [dA] | 20.5 [dA] | −2 [dB] | - | 19.9 [dA] | 11 [dA] | 19.8 [dA] | −20 [dB] |
| 160 | 6 [bc] | 22.6 [deA] | 8 [deA] | 20.6 [dB] | −2 [dB] | 1 [b] | 21.4 [eA] | 10 [eA] | 21.2 [eA] | −21 [eB] |
| 200 | 8 [d] | 24.1 [eA] | 7 [eA] | 21.5 [dB] | −2 [dB] | - | 21.5 [eA] | 10 [eA] | 21.5 [eA] | −22 [eB] |
| **Mean** | **6** | **20** | **11** | **18** | **−19** | **3** | **17** | **14** | **17** | **−17** |

Treatments with the same letter are not significantly different (*p* ≤ 0.05). The capital letters denote differences between Pn, Ps in P-plus and control treatments and the small letters denote difference between N rates; *n* = 320 for each location.

**Table 3.** P use indices for maize; N fertilizer rate; P fertilizer rate; Pn (uptake) and Ps (surplus) (kg·ha⁻¹); PUE (%).

| N Rate | Grabów | | | | | Baborówko | | | | |
|---|---|---|---|---|---|---|---|---|---|---|
| | P-Plus 35 kgP·ha⁻¹ | | | Control 0 kgP·ha⁻¹ | | P-Plus 35 kgP·ha⁻¹ | | | Control 0 kgP·ha⁻¹ | |
| | PUE | Pn | Ps | Pn | Ps | PUE | Pn | Ps | Pn | Ps |
| 0 | 11 [e] | 27.7 [aA] | 7 [aA] | 23.9 [aB] | −24 [aB] | 5 [c] | 17.6 [aA] | 17 [aA] | 16.0 [aA] | −16 [aB] |
| 50 | 12 [e] | 29.1 [abA] | 6 [abA] | 25.0 [abB] | −25 [abB] | 2 [a] | 21.8 [bA] | 13 [bA] | 21.0 [bA] | −22 [bB] |
| 100 | 4 [b] | 28.8 [abcA] | 6 [abA] | 27.3 [bA] | −27 [bB] | 3 [ab] | 24.8 [cA] | 10 [cA] | 23.6 [cA] | −23 [cB] |
| 150 | 1 [a] | 29.4 [abcA] | 6 [abA] | 28.9 [bcA] | −29 [bcB] | 3 [ab] | 26.6 [dA] | 8 [dA] | 25.5 [dA] | −26 [dB] |
| 200 | 5 [bc] | 30.2 [bcA] | 9 [bA] | 28.3 [bcB] | −28 [bcB] | 5 [c] | 27.2 [deA] | 8 [dA] | 26.8 [eA] | −27 [eB] |
| 250 | 5 [bc] | 29.9 [bA] | 6 [abA] | 28.2 [bcB] | −28 [bcB] | 3 [ab] | 27.7 [deA] | 7 [eA] | 27.6 [eA] | −28 [fB] |
| **Mean** | **7** | **29** | **7** | **27** | **−27** | **4** | **24** | **10** | **23** | **−23** |

Treatments with the same letter are not significantly different (*p* ≤ 0.05). The capital letters denote differences between Pn, Ps in P-plus and control treatments and the small letters denote difference between N rates; *n* = 320 for each location.

**Table 4.** P use indices for spring barley; N fertilizer rate; P fertilizer rate; Pn (uptake) and Ps (surplus) (kg·ha$^{-1}$); PUE (%).

| N Rate | Grabów | | | | | Baborówko | | | | |
| | P-Plus 31 kgP·ha$^{-1}$ | | | Control 0 kgP·ha$^{-1}$ | | P-Plus 31 kgP·ha$^{-1}$ | | | Control 0 kgP·ha$^{-1}$ | |
| | PUE | Pn | Ps | Pn | Ps | PUE | Pn | Ps | Pn | Ps |
|---|---|---|---|---|---|---|---|---|---|---|
| 0 | 3 [a] | 13.7 [aA] | 17 [aA] | 12.7 [aA] | −13 [aB] | 3 [b] | 6.9 [aA] | 24 [aA] | 6.0 [aA] | −6 [aB] |
| 30 | 6 [b] | 17.6 [bA] | 14 [bA] | 15.8 [bB] | −16 [bB] | 1 [a] | 10.5 [bA] | 21 [bA] | 10.2 [bA] | −10 [bB] |
| 60 | 4 [ab] | 20.4 [cA] | 11 [cA] | 19.2 [cA] | −19 [cB] | 2 [ab] | 14.4 [cA] | 17 [cA] | 13.8 [cA] | −14 [cB] |
| 90 | 7 [bc] | 22.0 [dA] | 9 [dA] | 19.9 [cdA] | −20 [cdB] | 3 [b] | 17.2 [dA] | 14 [dA] | 16.4 [dA] | −16 [dB] |
| 120 | 7 [bc] | 22.9 [dA] | 8 [dA] | 20.6 [deB] | −21 [deB] | 4 [bc] | 19.5 [eA] | 12 [eA] | 18.3 [eA] | −18 [eB] |
| 150 | 9 [c] | 24.0 [eA] | 7 [eA] | 21.3 [eB] | −2 [eB] | 3 [b] | 21.2 [fA] | 10 [fA] | 20.3 [fA] | −20 [fB] |
| **Mean** | **6** | **20** | **11** | **18** | **−18** | **3** | **15** | **16** | **14** | **−14** |

Treatments with the same letter are not significantly different ($p \leq 0.05$). The capital letters denote differences between Pn, Ps in P-plus and control treatments and the small letters denote difference between N rates; $n = 320$ for each location.

### 3.4. Effects of P Additions on Nitrogen Uptake and Use Efficiency

The N use indices calculated for winter oilseed rape, winter wheat, maize and spring barley for each location are presented in Tables 5–8.

Both N use efficiency and N uptake by all the crops exhibited higher values in Grabów compared to Baborówko. This left a greater N surplus in Baborówko (Tables 5–8). Although NUE and Yn in the control were slightly lower than the P-plus treatment, the impact of P management on these indices was not statistically significant. Consistently, withholding P additions in the control did not significantly affect the N surplus.

**Table 5.** N use indices for winter oilseed rape; N fertilizer rate, Yn (uptake), Ns (surplus) (kg·ha$^{-1}$); N use efficiency (NUE) (%) for the P fertilizer addition (P-plus) and control (no P addition).

| N Rate | Grabów | | | | | | Baborówko | | | | | |
| | P-Plus | | | Control | | | P-Plus | | | Control | | |
| | NUE | Yn | Ns | NUE | Yn | Ns | NUE | Yn | Ns | NUE | Yn | Ns |
|---|---|---|---|---|---|---|---|---|---|---|---|---|
| 50 | 145 [a] | 72 [a] | −22 [a] | 140 [a] | 70 [a] | −20 [a] | 91 [a] | 46 [a] | 4 [a] | 100 [a] | 50 [a] | 0 [a] |
| 100 | 100 [b] | 102 [b] | −2 [a] | 100 [b] | 101 [b] | −1 [a] | 73 [b] | 74 [b] | 26 [b] | 67 [b] | 70 [ab] | 30 [b] |
| 150 | 80 [ab] | 120 [bc] | 30 [b] | 78 [c] | 117 [bc] | 33 [b] | 65 [c] | 98 [c] | 52 [c] | 59 [c] | 89 [c] | 61 [c] |
| 200 | 70 [b] | 140 [cd] | 60 [c] | 67 [cd] | 134 [c] | 66 [c] | 54 [d] | 107 [c] | 93 [d] | 52 [cd] | 104 [d] | 96 [d] |
| 250 | 62 [b] | 155 [d] | 95 [d] | 57 [d] | 141 [c] | 109 [d] | 47 [d] | 118 [d] | 132 [e] | 46 [d] | 114 [e] | 136 [e] |
| **Mean** | **91** | **118** | **32** | **88** | **112** | **37** | **66** | **89** | **61** | **65** | **86** | **81** |

The small letters denote difference between NUE, Yn, Ns in P-plus and control treatments in the same location; Treatments with the same letter are not significantly different ($p \leq 0.05$); $n = 280$ for Grabów; $n = 240$ for Baborówko.

**Table 6.** N use indices for winter wheat; N fertilizer rate, Yn (uptake), Ns (surplus) (kg·ha$^{-1}$); NUE (%) for the P fertilizer addition (P-plus) and control (no P addition).

| N Rate | Grabów | | | | | | Baborówko | | | | | |
| | P-Plus | | | Control | | | P-Plus | | | Control | | |
| | NUE | Yn | Ns | NUE | Yn | Ns | NUE | Yn | Ns | NUE | Yn | Ns |
|---|---|---|---|---|---|---|---|---|---|---|---|---|
| 40 | 196 [a] | 78 [a] | −38 [a] | 193 [a] | 77 [a] | −37 [a] | 155 [a] | 62 [a] | −22 [a] | 153 [a] | 61 [a] | −21 [a] |
| 80 | 134 [b] | 107 [b] | −27 [b] | 131 [b] | 105 [b] | −25 [b] | 111 [b] | 89 [b] | −9 [b] | 106 [b] | 85 [b] | −5 [b] |
| 120 | 106 [c] | 128 [c] | −8 [c] | 103 [c] | 123 [c] | −3b [c] | 89 [c] | 107 [c] | 13 [c] | 89 [c] | 107 [c] | 13 [c] |
| 160 | 92 [d] | 147 [d] | 13 [d] | 91 [cd] | 146 [d] | 14 [d] | 78 [c] | 124 [d] | 36 [d] | 76 [cd] | 122 [d] | 38 [d] |
| 200 | 83 [d] | 166 [e] | 34 [e] | 81 [d] | 162 [e] | 38 [e] | 65 [d] | 130 [d] | 69 [e] | 64 [d] | 128 [d] | 72 [e] |
| **Mean** | **122** | **125** | **−5** | **120** | **123** | **−3** | **100** | **102** | **17** | **98** | **101** | **19** |

The small letters denote difference between NUE, Yn, Ns in P-plus and control treatments in the same location; Treatments with the same letter are not significantly different ($p \leq 0.05$); $n = 320$ for each location.

**Table 7.** N use indices for maize; N fertilizer rate, Yn (uptake), Ns (surplus) (kg·ha$^{-1}$); NUE (%) for the P fertilizer addition (P-plus) and control (no P addition).

| N Rate | Grabów | | | | | | Baborówko | | | | | |
|---|---|---|---|---|---|---|---|---|---|---|---|---|
| | P-Plus | | | Control | | | P-Plus | | | Control | | |
| | NUE | Yn | Ns | NUE | Yn | Ns | NUE | Yn | Ns | NUE | Yn | Ns |
| 50 | 231 [a] | 115 [a] | −65 [a] | 227 [a] | 114 | −64 [a] | 195 [a] | 97 [a] | −47 [a] | 199 [a] | 99 [a] | −49 [a] |
| 100 | 154 [b] | 154 [b] | −54 [a] | 145 [b] | 145 | −45 [b] | 145 [b] | 145 [b] | −45 [a] | 138 [b] | 138 [b] | −38 [a] |
| 150 | 107 [c] | 161 [b] | −11 [b] | 99 [c] | 148 | 2 [c] | 112 [c] | 168 [c] | −18 [b] | 104 [c] | 156 [c] | −6 [b] |
| 200 | 89 [d] | 179 [c] | 21 [c] | 85 [cd] | 170 | 30 [d] | 87 [d] | 175 [cd] | 25 [c] | 87 [d] | 175 [d] | 25 [c] |
| 250 | 71 [e] | 175 [c] | 75 [d] | 69 [d] | 173 | 77 [e] | 74 [e] | 185 [d] | 65 [d] | 76 [d] | 191 [e] | 59 [d] |
| **Mean** | **130** | **156** | **−7** | **125** | **150** | **0** | **123** | **154** | **−4** | **121** | **151** | **−2** |

The small letters denote difference between NUE, Yn, Ns in P-plus and control treatments in the same location; Treatments with the same letter are not significantly different ($p \leq 0.05$); $n = 320$ for each location.

**Table 8.** N use indices for spring barley; N fertilizer rate, Yn (uptake), Ns (surplus) (kg·ha$^{-1}$); NUE (%) for the P fertilizer addition (P-plus) and control (no P addition).

| N Rate | Grabów | | | | | | Baborówko | | | | | |
|---|---|---|---|---|---|---|---|---|---|---|---|---|
| | P-Plus | | | Control | | | P-Plus | | | Control | | |
| | NUE | Yn | Ns | NUE | Yn | Ns | NUE | Yn | Ns | NUE | Yn | Ns |
| 30 | 211 [a] | 63 [a] | −33 [a] | 202 [a] | 61 [a] | −31 [a] | 132 [a] | 39 [a] | −9 [a] | 135 [a] | 41 [a] | −9 [a] |
| 60 | 136 [b] | 81 [b] | −21 [b] | 132 [b] | 78 [b] | −19 [b] | 101 [b] | 61 [b] | −1 [b] | 103 [b] | 62 [b] | −2 [b] |
| 90 | 114 [c] | 102 [c] | −12 [c] | 106 [c] | 95 [c] | −5 [c] | 89 [ab] | 81 [c] | 9 [c] | 82 [c] | 74 [c] | 16 [c] |
| 120 | 99 [d] | 119 [d] | 1 [d] | 96 [c] | 114 [d] | 6 [d] | 77 [bc] | 92 [d] | 28 [d] | 76 [cd] | 91 [d] | 29 [d] |
| 150 | 88 [d] | 132 [e] | 18 [e] | 81 [d] | 121 [d] | 29 [e] | 70 [cd] | 105 [e] | 45 [e] | 66 [d] | 99 [e] | 51 [e] |
| **Mean** | **130** | **99** | **−10** | **123** | **94** | **−4** | **94** | **76** | **14** | **92** | **73** | **17** |

The small letters denote difference between NUE, Yn, Ns in P-plus and control treatments in the same location; Treatments with the same letter are not significantly different ($p \leq 0.05$); $n = 320$ for each location.

Maize exhibited the highest values of NUE in the range of 71%–231%, with an average of 127%. The average N uptake by maize in both locations was greater than the minimal EU NEP [40] value of 80 kg N·ha$^{-1}$ (Table 7). The crop removed the greatest quantity of soil N in the range of 50–150 kg N·ha$^{-1}$ applied in the experiment (43 kg·ha$^{-1}$ on average). Over the entire range of N fertilizer rates, winter oilseed rape showed the lowest indices of NUE (46–145%), N uptake (50–155 kg N·ha$^{-1}$) and there about 53 kg N·ha$^{-1}$ remained in the soil (Table 5). The average value of NUE calculated for winter wheat was 110% on average, and the mean N uptake was 114 kg N·ha$^{-1}$. In Grabów, the NUE for barley was higher than Baborówko and oscillated over the ranges 88–211 kg N·ha$^{-1}$ and 66–135 kg N·ha$^{-1}$, respectively. The uptake of N by barley in Baborówko was smaller, thus the average value of surplus N in this location in the control P treatment was positive.

*3.5. Changes in Available P in the Soil*

In Grabów, the average content of available P in soil in the P-plus treatment increased from the initial 2003 value of 69.8 to 90.3 mg P·kg$^{-1}$ soil between 2007 and 2018. In Baborówko, a similar increase was observed—from 111.3 to 134.4 mg P·kg$^{-1}$ soil. Through the process of P soil mining in the control treatment (no P added), the average concentration dropped to 63.3 mg P·kg$^{-1}$ soil in Grabów and to 107.8 mg P·kg$^{-1}$ soil in Baborówko. An analysis of variance showed a significant reduction in available P in soil at increasing rates of N fertilizer addition in Grabów as a result of the higher crop yields and P uptake. In Grabów, soil P levels decreased in both the P-plus and control as N doses increased reaching as low as 51.5 mg P·kg$^{-1}$ soil (Figure 9a). In contrast, this trend was not seen in Baborówko (Figure 9b) which had much higher soil P levels than Grabów, even though the P content of the soil was also about 20% higher in the P-plus treatment than the control.

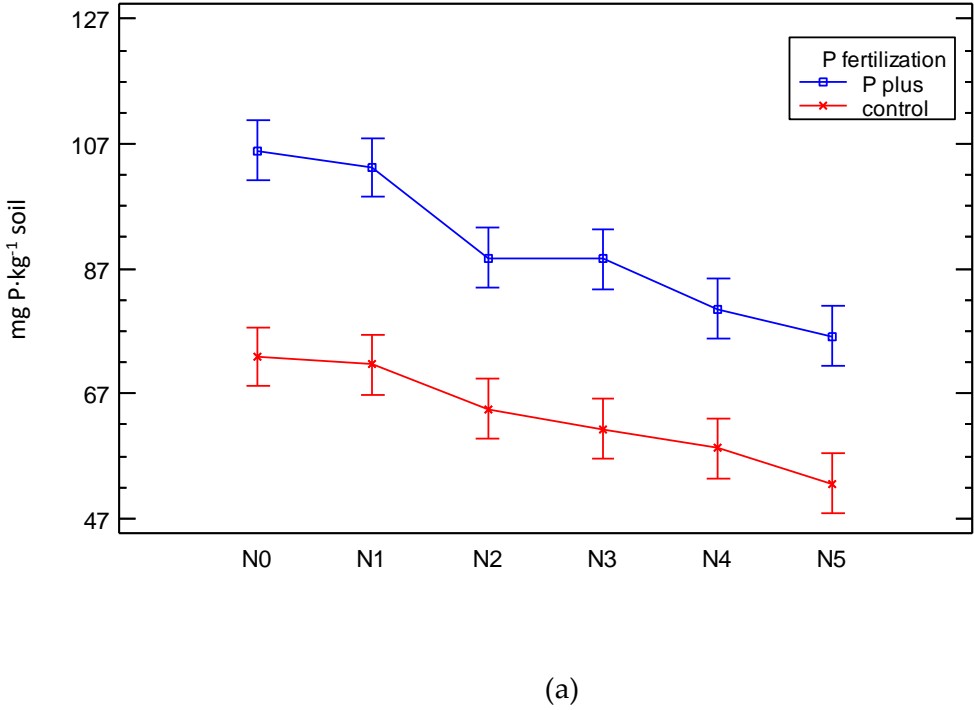

(a)

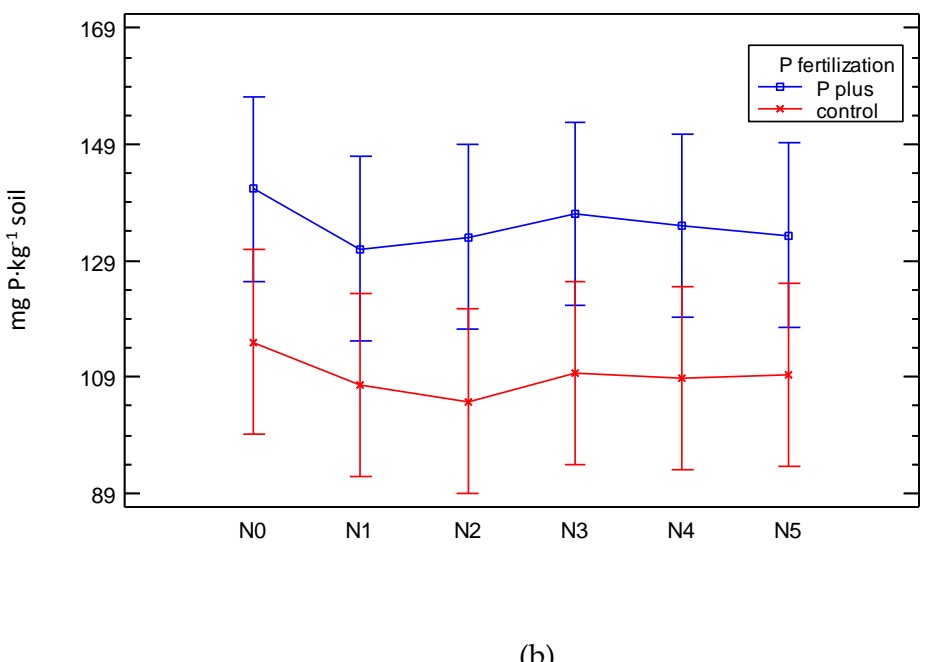

(b)

**Figure 9.** Running averages of P content in the soil in Grabów (**a**) and Baborówko (**b**) (2007–2018); 95% Tukey HSD intervals; *n* = 288 for each location (P fertilizer Grabów HSD 4.026, Baborówko HSD 13.32).

## 4. Discussion

The 16-year experiment, the results of which have been presented here, was conducted on soils with high (Grabów) and very high (Baborówko) contents of available P in the plough layer. Such soils should be considered as typical for Poland, as they account for almost 60% of the country. The achieved soil acidity in Grabów was slightly acid (pH $K_{Cl}$ 6.2) and neutral in Baborówko (pH $K_{Cl}$ 6.8). The Polish recommendation system for P fertilizer application is based on soil P supply and on crop P demand.



Fertilizer recommendations using five classes of available P (determined by the Egner-Riehm DL method in topsoil) have been identified: very low, low, medium, high and very high. It is recommended to achieve at least a medium level of soil P (44.1–65.4 mg·kg$^{-1}$ soil). In the experiments, the soil was characterized by a low content of soil organic carbon (5.0 g org C·kg$^{-1}$ soil) and no exogenous organic matter has been applied for over 30 years.

The results of the experiments confirm the hypothesis that several years without P fertilizer addition did not exhaust the available soil P to levels that limit crop productivity. In Grabów, where the initial value of available soil P was 69.8 mg P·kg$^{-1}$ soil, annual applications of P fertilizer led to an increase in available soil P over the whole range of N fertilizer rates, although the effect was suppressed at higher N levels. The P level in the soil remained within the high class of P content.

After P fertilizer addition was stopped in 2003, the topsoil P in the controls significantly dropped. It should be noted that this decrease in soil P was further enhanced by increases in N fertilizer rates. At no or low additions of N fertilizer, N0-N1 (0–50 kg N·ha$^{-1}$), the content of available P in soil was close to the initial value of 69.8 mg P·kg$^{-1}$ soil. In the range of medium rates, N2-N3 (60–120 kg N·kg$^{-1}$) the P concentration was 63.2 on average. At the highest N4-N5 rates (120–200 kg N·kg$^{-1}$), the content dropped further to 57.5 and 52.4 mg P·kg$^{-1}$ soil, respectively. Going beyond the highest N5 rate could result in P limitation of crop productivity.

In Baborówko, where the experiment was located on soil with a very high content of available soil P (111.8 mg P·kg$^{-1}$), the soil P content remained high and no interaction between the added N and P fertilizer was noted. Similarly, the P-plus treatment only increased available soil P to 133 mg P·kg$^{-1}$. Crop yields were lower than in Grabów, and the lowest values of P uptake were noted. Hence, the P balance in Baborówko showed fewer negative values, and the N additions did not affect the soil P content.

Phosphorus in soil occurs in inorganic (Pi) and organic (Po) forms, which undergo various transformations, affecting their fate in soil and their bioavailability to plants [42,43]. The share of Pi in the total P (Pt) pool is estimated at 35% to 70% [44]. The soil solution contains only a small fraction of orthophosphate ions available to plants, which during plant growth are replenished from mobile reserves of Pi and organic matter. The remainder of Pt is the so-called "spare" P, the release of which is very slow [45,46]. Nevertheless, P is not irreversibly fixed in soil and is frequently mobilized during the growing period [47]. The mineralization of Po occurs under the influence of enzymes secreted by bacteria, fungi and plant roots. The rate of mineralization is influenced by soil environmental conditions, primarily pH and temperature [48,49]. In acid soils, P is fixed by free oxides and hydroxides of aluminum and iron. In alkaline soils it is greatly adsorbed onto lime surfaces and clay matter and precipitated in the form of different Ca-P minerals. All these fixation reactions contribute to reducing the solubility and plant availability of applied P [50,51]. In our experiments with the relatively high levels of soil P, plant availability was found to be optimal and had no constraints due to this soil parameter for P uptake being observed.

In Grabów, the P additions resulted in lower crop uptake, particularly at the highest N fertilizer rates. In Baborówko, where all the crops had lower yields, there were no significant differences between the P-plus and control treatments. The sixteen years of soil P mining in the zero-added P controls caused a negative P surplus. The average P surplus calculated for one rotation in Grabów was −82 kg P·ha$^{-1}$ and it was −68 kg P·ha$^{-1,}$ in Baborówko. In effect then, over the sixteen years (four rotations), the capacity to reduce soil P was −328 and −272 kg P·ha$^{-1}$, respectively. Nevertheless, such a negative balance has not affected the crop productivity in Baborówko, even at the highest rates of applied N fertilizer. This is a significant finding for managing P levels in agriculture soil. Soil mining by crops receiving only N fertilizer could have a major impact on reducing legacy P (from many decades of overfertilization) in the Baltic Sea drainage basin.

As reported in the literature, the P fertilizer inputs in the world are two to five times greater than the amount exported in the final products [52]. It was thought to be of great importance in the early years in the mid-19th century to apply more P than was removed in the harvested crop

to achieve an acceptable yield. This concept led to the accumulation of large quantities of surplus P fixed in agriculture soils [53]. Johnson and Syers [54] found in experiments conducted between 1856 and 1901 in Rothamsted, that P fertilizers were applied year to year and the resulting P balance was positive. However, following 100 years after the last P application, residual P was still being recovered in winter wheat compared to treatments where additional P had been applied since 1856. Phosphorus accumulation in soils typically occurs when P fertilizer is added, whilst P depletion can occur under intensive cropping unless there is an external supply [55]. Sun et al. [56] reported that over a 24-year experiment with winter wheat and maize, despite P depletion, yields were maintained, albeit at low values.

In our experiments, the rate at which P has been released in soil each year has been sufficient for several years to provide enough P to produce the optimum yield for the modern high-demand for both N and P cultivars. The uptake of P by the crops was comparable between the P-plus and control treatments. The P use efficiency calculated as the difference between P uptake in plants fertilized with P and not fertilized with P was low and achieved at most 12%.

The data presented by Lasaletta et al. [57] show that currently only 47% of the reactive N added globally onto cropland is converted into harvested products, compared to 68% in the early 1960s. This means, that more than half of the N used in crop fertilizer is currently lost to the environment. The prime cause of N loss is through nitrate leaching or denitrification in connection with excessive rainfall. When the whole dose of N fertilizer is applied prior to planting, a significant risk of losses during the spring occurs. The time between N application and its active absorption by the crops, provides numerous opportunities for N leaching, clay fixation, immobilization, denitrification and volatilization [58]. Developing fertilizer strategies that increase NUE could reduce unnecessary input costs to farmers and environmental impact of N losses. Examples include adjusting N fertilizer additions based on knowledge about the soil mineral forms of N, phasing N fertilizer applications or using fertilizers with a nitrification inhibitor [59].

Nitrogen use efficiency is suggested to be a key indicator for agricultural systems. Several studies have estimated NUE in cropping systems, but these studies often use different definitions, input data and assumptions [60]. In this paper, NUE indices were calculated according to the method proposed by the EU NEP [40], using three related indicators: NUE, N surplus and N uptake. The impact of P additions on NUE indices for the most common crops in Poland was examined. Contrary to what might be expected for such a long period, the effect of increased soil P mining (by the crops) on the NUE indices was negligible. This means that P mining did not increase the risk of N losses due to insufficient N uptake by crops. Higher values of NUE were noted in Grabów compared to Baborówko due to a greater crop yield. Nitrogen uptake was enhanced in Grabów due to the more favorable rainfall conditions through the vegetation periods. Maize exhibited the highest values of NUE and N uptake and similarly the crop gave the lowest N surplus (average minus 7 kg N per ha). Over the entire range of N fertilizer additions, winter oilseed rape showed the lowest NUE and Yn values and left a soil surplus of about 50 kg N per ha per year. The cultivation of winter oilseed rape is problematic for the reason of considerable N surplus [61,62]; simultaneously the crop requires a high quantity of P for optimum growth [63,64] and P deficiency restricts both top and root growth [65,66]. In the presented experiments, surplus N has not been affected by long-term P fertilizer additions, although in Baborówko, under the rates of 200 and 250 kg N·ha$^{-1}$, the surplus exceeded the desired value recommended by the EU NEP.

According to the results obtained by Quemada et al. [67], based on the data collected through surveys of 195 arable farms in Europe, with the prevalence of cereals grown in 2–3 crop rotations, the mean value of NUE was 60%. This value was obtained for a mean N input of 176 kg N·ha$^{-1}$ and 105 kg N·ha$^{-1}$ output, resulting in an N surplus of ca. 72 kg N·ha$^{-1}$. In our experiments with inputs close to 170 kg N·ha$^{-1}$—i.e., 150 kg N·ha$^{-1}$ for winter oilseed rape, maize and barley—the mean NUE was 76%, N uptake was 120 kg N·ha$^{-1}$ and N surplus was 30 kg N·ha$^{-1}$. Similarly, at an input rate of 160 kg N·ha$^{-1}$ for barley, the indices were NUE 84%, Yn 134 kg N·ha$^{-1}$ and Ns 25 kg N·ha$^{-1}$.

The relatively high values of NUE and Yn in our experiments might be the result of the split application of N fertilizers.

In Poland, the average N fertilizer rates in the production of cereals over recent years do not exceed 90 kg N·ha$^{-1}$ per year and for maize and winter oilseed rape do not exceed 145 kg N·ha$^{-1}$·year$^{-1}$ [68]. Considering that the annual consumption of mineral fertilizers between 2010 and 2018 oscillated around 130 kg NPK·ha$^{-1}$ (of which 75 kg was N [69]) on soil with an average content of 73.6 mg P·ha$^{-1}$ prevailing in the country, it might be recommended to forsake year-to-year P applications to avoid its further accumulation in the soil. Nevertheless, it is important to gauge the addition of P fertilizers to the considered N fertilizer rates since the high N uptake intensifies soil P mining/depletion. It should be emphasized, however, that these recommendations concern soils with optimal conditions and where acidity does not limit P uptake. Moreover, soil testing for available P in topsoil is critical for assessing the environmental consequences of unbalanced N and P applications. This highlights the importance of soil testing prior to making decisions on fertilizer use [70,71].

## 5. Conclusions

Long-term N and P fertilizer experiments over 16 years with winter oilseed rape, winter wheat, maize and spring barley revealed the presence of significant soil P mining causing a reduction in the content of available forms of P but without negative impacts on crop productivity. In the Grabów region, where the initial value of available P (Egner-Riehm) was classified as high, 69.8 mg P·kg$^{-1}$ soil, the sixteen years of P soil mining led to a decrease in available topsoil P to an average of 63.2 mg P·kg$^{-1}$ soil. This dropped the soil P classification level to medium. Additionally, there was a close relation between the intensity of soil P depletion and the range of applied N fertilizer rates. The smaller N doses of 30–50 kg N·ha$^{-1}$ did not change the soil P levels over the long term but the highest rates, 150–250 kg N·ha$^{-1}$, reduced available soil P by about 35%. This points to the need for monitoring P levels in soil, especially when intensified N fertilizer applications are considered. In contrast, in the Baborówko region, where soil P content was very high (111.2 mg P·kg$^{-1}$ soil), the added N fertilizer did not affect the changes in soil P over the long term. Moreover, the year-to-year addition of P fertilizer resulted in an unnecessary accumulation of available P in the soil.

Increased grain yields, higher N use efficiency (NUE) and higher N uptake by all the crops were noted in Grabów but not in Baborówko where soil P levels were much higher to start with. At the relatively high levels of available soil P in these experiments, additional P fertilizer did not affect the NUE indices. Nitrogen use efficiency for each crop dropped systematically with increased N doses and the mean NUE was 127% for maize, 111% for winter wheat and 110% for winter oilseed rape and barley but P fertilizer additions had no impact. Therefore, due to economic and environmental concerns it can be recommended to refrain from year-to-year P fertilizer applications to soils that already have high or very high P soil content. This can be continued for several years (at least 16 in this case). At the same time, this deprivation of P fertilizer will have no significant negative impacts on crop yields. Still, over the long term, P fertilizer regimes should be closely linked to N fertilizer strategies to retain optimal conditions for productivity. It is also very important that the negative P balance surplus at the field level significantly increases the utilization of P from soil resources by plants and reduces the phosphorus pool potentially at risk of leaching to ground and surface waters. Limiting phosphorus fertilization is a measure recommended by HELCOM and aimed at controlling the P load introduced in the Baltic Sea. In the long-term perspective, it could contribute to the achievement of the reduction targets set out in the Baltic Sea Action Plan. The implications of this work on reducing soil P surpluses and runoff losses to the Baltic Sea drainage basin and other receiving water systems around the world are significant.

**Author Contributions:** Conceptualization, A.R.; Methodology, A.R.; Validation, A.R.; Formal analysis, A.R.; Investigation, A.R.; Resources, A.R.; Data curation, A.R.; Writing—Original Draft Preparation, A.R., P.S.; Writing—Review and Editing, A.R., P.S.; Visualization, A.R., P.S.; Supervision, A.R.; All authors have read and agreed to the published version of the manuscript.

**Funding:** This research received no external funding

**Conflicts of Interest:** The authors declare no conflict of interest.

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
