# Peer review of "Productive and Environmental Consequences of Sixteen Years of Unbalanced Fertilization with Nitrogen and Phosphorus—Trials in Poland with Oilseed Rape, Wheat, Maize and Barley"

_agronomy, doi:10.3390/agronomy10111747_

Round 1

Reviewer 1 Report

Review

Agronomy 957769

The Authors have investigated an interesting topic – effect of unbalanced fertilization with nitrogen and phosphorus on crop yield in a long term experiment.  In general, the organization, contents and the structure of the article are satisfactory. However, some parts of the manuscript need to be corrected.

  • The introduction provides sufficient background, however the according to the manuscript title there should be more about nitrogen. Currently, the article is more about phosphorus than nitrogen.
  • The authors did not present a clear research hypothesis and the purpose of manuscript.
  • The Materials and Methods chapter is incomplete and not very informative (it mainly concerns the form of presentation). Was a Variance analysis carried out ? Data on the content of plant available K, Mg (and others) should be presented separately for each locality. What methods and laboratory equipment were used to analyze the N and P content? Source reference should be added to the Egner-Riehm method. The soil classification according to WRB must be provided.
  • The subsection titles in Results chapter are too long; e.g. instead “ The impact of phosphorus …..” enough is “Nitrogen management indices”
  • The word either "fertilization" or "fertilization" should be used consistently.
  • Consistency is needed throughout the manuscript with the use of abbreviations, as well as chemical symbols; e.g. Page 1, line 35; or Page 6, line 183-184.
  • Page 6, line 188-189, Table 1 - there is no information what the acronym Nup means, I understand it's Nitrogen uptake? An explanation of the abbreviation should be in this subsection, or in Materials and Methods
  • Page 6-7, in Tables 1-4 the Ns values do not match when the algorithm for calculating the Nitrogen surplus is taken into account (Ns = F – Yn). Why ?
  • P2O5 / 100g units should be converted to mg P kg-1 of soil
  • Page 3, line 103 shouldn't be “Nitrogen uptake efficiency = NUpE” ?
  • Figure 1b, is there Grabow locality, but school be Baborowko.
  • The uptake of phosphorus by plants is very laconic. Why the P balance was not calculated in a similar way as for N?
  • There is no information whether the effect of the doses of N was significant on NUE, Nup and Ns
  • The discussion is poorly written. The order should be reordered, eg start with the sentence Page 9, Line 239. Then consistently explain the differences in soil reaction in the two locations. Line 256-261 and 269-281 very general discussion. Why was the difference between soil and in Boborowko Grabow in terms of changes in concentration of P after application of N fertilizers.
  • Conclusions should be corrected. They should be more related to the research carried out and the results obtained. The results concerned different soils.
  • The word either "fertilization" or "fertilization" should be used consistently.
  • Consistency is needed throughout the manuscript with the use of abbreviations, as well as chemical symbols; e.g. Page 1, line 35; or Page 6, line 183-184.
  • Page 6, line 188-189, Table 1 - there is no information what the acronym Nup means, I understand it's Nitrogen uptake? An explanation of the abbreviation should be in this subsection, or in Materials and Methods.
  • Page 6-7, in Tables 1-4 the Ns values do not match when the algorithm for calculating the Nitrogen surplus is taken into account (Ns = F – Yn). Why ?
  • Page 288, line 288 - the purpose of demonstrating the number of 850 kg N / ha is incomprehensible?

Author Response

Dear Sir or Madam

I send you my response to your comments in the attached file.

Yours Sincerely

Agnieszka Rutkowska

Reviewer 2 Report

The manuscript entitled “The productive and environmental consequences of the sixteen years of unbalanced fertilization with nitrogen and phosphorus” describes the effect of P and N fertilization on yield of four crops (oilseed rape, winter wheat, maize and spring barley) in Poland. Nitrogen use efficiency (NUE) for each crop was also reported.
The study is a long-term study (16 years) that, for this reason, could have a great potential as source of precious and novel information on cropping system fertilization.
Unfortunately, the manuscript was poorly written. There are several comments in almost each section that need to be addressed and aspects that need to be clarified.
Overall, major revisions are required before acceptance.
The major issues are listed below.
Abstract: Check the unit of measurement of P fertilizer applied (lines 15 and 16), probably it is kg ha-1 and not mg 100 g-1 soil
Introduction: In my opinion this section does not provide a sufficient background. Considering that this study aims to explore the effect of P and N fertilization on performance of four crops in a long period, very few citations were listed by the authors. Moreover, the introduction seems to be focused exclusively on P, while the role of N is neglected.
Materials and Methods: Pivotal information are lacking. What level of P fertilization P minus correspond to? It was stated in the abstract but not in this section! How large the plots were? How many plants/seeds were collected for each plot to determine yield? How do you determine yield for each crop?
Lines 79-80 All crops must be identified with the name and in brackets Genus species and patronymic (Triticum aestivum L,) for the winter oilseed rape (Brassica napus L. var. oleifera).
Lines 85 why is pHKCl reported and not pHH2O ?
Lines 92. The difference between P-plus and P-minus is not clear
Results: Too short for a complex and long study like this. In the paragraph 3.1 , the authors commented very shortly just the last four years of the study without analyzing the 16-year trend for each crop. Moreover, I can not understand where the data reported from line 145 to 148 are shown. Are they “not shown” data? In this case it must be stated!
Fig. 1b Grain yields of oilseed rape in Baborówko not in Grabów
Fig. .. In the figures showing statistically different averages, the L.S.D. must be shown.
Discussion: this section must be improved as previously suggested for the introduction and results section

Author Response

(The authors gave the same response as above.)

Reviewer 3 Report

There is necessary to make a correction (leaving out of pause) in words "sufficient, optimum, availability, between" in rows No 57, 58, 61 and 64.

Two replications in experiment seems to be little - for statistical valuation three replications are minimum, four replications are optimum. On the other hand, this negative factor is compensated by long-term period of experiments (since 2003), so I recommend to use all data for publication. But will be fine a little bit better describe why do you have only two replications.

Author Response

(The authors gave the same response as above.)

Round 2

Reviewer 1 Report

Manuscript agronomy-957769 has been significantly improved. Please just correct the following parts of the text:

Line 14. Treatment instead of variant

Line 18. Abstract - please enter the full name of the NUE, next the abbreviation in brackets

Line 24. I suggest changes to keywords: crops productivity, nitrogen use effectivity, nutrient balance, P recovery efficiency, plant available P . In general, words should not be repeated in the title.

Line 160-174. In the equations, please clearly identify whether variables Yn or Pn represent the total nutrient uptake (e.g. total N uptake), and whether this is equivalent to removing components from the field.

Use the symbols N and P more often in the text, e.g. Line 28, 32, 39, et cetera

Supplementary material: in table is LSD, not HSD (Tukey’s test). It should be changed.

Author Response

Dear Sir or Madam

The cover letter has been attached.

Yours sincerely

Agnieszka Rutkowska

Reviewer 2 Report

The manuscript entitled "The productive and environmental consequences of the sixteen years of unbalanced fertilisation with nitrogen and phosphorus" was improved from the first version.

- I just suggest that the authors support what they stated in lines 82-84 ("More appropriate management of N use is therefore of key importance, but it should be  emphasised that the proper management of the element is linked to the balanced fertilisation with other nutrients to achieve both the economic and environmental goals".) with some references. I can suggest the following one: Rossini, F.; Provenzano, M.E.; Sestili, F.; Ruggeri, R. Synergistic Effect of Sulfur and Nitrogen in the Organic and Mineral Fertilization of Durum Wheat: Grain Yield and Quality Traits in the Mediterranean Environment. Agronomy 20188, 189; but you can add others too.  

- The name of place in Figure 1b is still incorrect  

Author Response

(The authors gave the same response as above.)
